# Perceptions of the Conditions and Barriers in Implementing the Patient Blood Management Standard by Anesthesiologists and Surgeons

**DOI:** 10.3390/healthcare12070760

**Published:** 2024-03-31

**Authors:** Andrea Kazamer, Radu Ilinca, Iulia-Ioana Stanescu-Spinu, Dan Adrian Lutescu, Maria Greabu, Daniela Miricescu, Anca Magdalena Coricovac, Daniela Ionescu

**Affiliations:** 1Department of Anaesthesia and Intensive Care I, Iuliu Hatieganu University of Medicine and Pharmacy, 8 Victor Babes Street, 400347 Cluj-Napoca, Romania; daniela_ionescu@umfcluj.ro; 2CREST Association, 48 Alexandru Odobescu Street, 440069 Satu Mare, Romania; 3Discipline of Medical Informatics and Biostatistics, Faculty of Dentistry, Carol Davila University of Medicine and Pharmacy, 4–6 Eforie Street, 020021 Bucharest, Romania; dan-adrian.lutescu@drd.umfcd.ro; 4Discipline of Physiology, Faculty of Dentistry, Carol Davila University of Medicine and Pharmacy, 8 Eroii Sanitari Blvd, 050474 Bucharest, Romania; iulia.stanescu@umfcd.ro; 5Discipline of Biochemistry, Faculty of Dentistry, Carol Davila University of Medicine and Pharmacy, 8 Eroii Sanitari Blvd, 050474 Bucharest, Romania; maria.greabu@umfcd.ro (M.G.); daniela.miricescu@umfcd.ro (D.M.); 6Discipline of Embryology, Faculty of Dentistry, Carol Davila University of Medicine and Pharmacy, 8 Eroii Sanitari Blvd, 050474 Bucharest, Romania; anca.coricovac@umfcd.ro; 7Outcome Research Consortium, Cleveland, OH 44195, USA

**Keywords:** patient blood management, patient safety, patient-blood-management standard implementation, anemic patients

## Abstract

Patient Blood Management (PBM) as a multidisciplinary practice and a standard of care for the anemic surgical patient is playing an increasingly important role in reducing transfusions and optimizing both clinical outcomes and costs. The success of PBM implementation depends on staff awareness and involvement in this approach. The main objective of our study was to explore physicians’ perceptions of the conditions for implementing PBM in hospitals and the main obstacles they face in detecting and treating anemic patients undergoing elective surgery. This cross-sectional descriptive study includes 113 Romanian health units, representing 23% of health units with surgical wards nationwide. A 12-item questionnaire was distributed to the participants in electronic format. A total of 413 questionnaires representing the perceptions of 347 surgeons and 66 anesthesia and intensive-care specialists were analyzed. Although a lack of human resources was indicated by 23.70% of respondents as the main reason for not adhering the guidelines, the receptiveness of medical staff to implementing the PBM standard was almost 90%. In order to increase adherence to the standard, additional involvement of anesthesia and intensive-care physicians would be necessary from the perception of 35.70% of the responders: 23.60% of surgeons and 18.40% of hematologists.

## 1. Introduction

The Patient Blood Management (PBM) standard represents the integrated use of appropriate transfusion practices, multidisciplinary collaboration, preoperative diagnosis of anemia and appropriate clinical decision-making using evidence-based guidelines [1,2]. Anemia as a risk factor appears in about 29% of surgical patients. Transfusion, as one of the most important modalities of treating anemia, also carries risks [3].

Despite the fact that the new hemoglobin-based oxygen carriers (novel HBOC molecules including polymerization and biomimetic strategies and the new encapsulated molecules) are extensively studied and may potentially provide oxygenation when red blood cells (RBCs) are not available, and are undergoing extensive research for clinical evaluation and to overcome logistical barriers [4,5], management of perioperative anemia and transfusion remains the main approach in the context of PBM.

Recent publications found that implementation of a multimodal PBM program (using the three pillars) resulted in a 39% reduction in transfusion rates, in addition to reductions in hospital length of stay and an overall 11% reduction in mortality rates [6]. In addition, the reduction in adverse events and complications associated with transfusions, thus improving patient safety, and saving costs in patient care are strong arguments in favor of implementing PBM [7]. The three pillars of PBM are centered on the patient’s condition and disease process, aiming to avoid unnecessary transfusion. Specific PBM measures include individualized assessment of both anemia and thrombotic risk, administration of tranexamic acid, intravenous iron administration, supplementation with folic acid and vitamin B12 to enhance RBCs production, minimized blood loss and introduction of transfusion restriction levels, etc. Monitoring the implementation of measures through clinical audit with a focus on transfusion rates can lead to improved practice by motivating stakeholders in patient care [8].

However, underlying the advantages of PBM is not sufficient for ensuring its implementation in everyday practice and integration into organizational culture. For this, different strategies have been proposed in the literature, starting with update meetings, multidisciplinary staff meetings [9] and knowing the institutional and literature statistics on the advantages of PBM. These strategies recommended by the literature resulted in an increase from 27.40% to 36.71% of patients treated for anemia before surgery. Nevertheless, only 14.29% of patients who were treated for anemia in 2022 were given iron at least 7 days before surgery, as recommended by the PBM standard [10].

Thus, from a day-to-day perspective, hospitals still do not meet the necessary conditions for PBM measures to be integrated into daily clinical practice. It has been shown that such conditions include the fact that some hospitals did not add PBM to their transfusion committees tasks, or that there are organizational obstacles involving clinical pathways, interdisciplinary communication, or that there is still a lack of interest in and knowledge of PBM [11,12].

Hence, to achieve the goals of implementing the PBM standard, specific local/institutional barriers need to be identified and addressed accordingly [10]. Health professionals may benefit from more explicit guidance on implementation strategies if these measures were best suited to their context and an implementation model existed to help identify barriers and to understand what works and what drives variation in practice [6].

Most studies in the field have focused on assessing the impact of implementing the PBM standard and defining strategies to support integrated implementation. However, specific barriers still need to be identified and strategies to encourage integrated implementation of the PBM standard need to be anchored in the context of organizational culture and in overcoming the specific barriers identified [7].

In this context, in our study we aimed to identify the barriers that hinder the implementation of the PBM standard and those conditions that support the use of specific measures recommended by PBM in the daily work of surgeons and anesthesiologists.

## 2. Materials and Methods

This descriptive cross-sectional study included 113 Romanian hospitals, representing 23% of the health units with surgical departments in our country. The study consisted of a questionnaire addressed to medical staff. The questionnaire used was based on research on PBM implementation strategies. There were 12 items on the questionnaire, divided into 3 categories. The first category inquired about the respondents’ opinions of the prerequisites that need to be met for hospitals to adopt the PBM standard. The perception of the obstacles to PBM implementation was the subject of the second set of questions. The third category was devoted to staff perceptions of the level of PBM standard implementation. Five experts in PBM standard implementation pretested and reviewed the prepared questionnaire for accuracy and clarity. Following the recommendations, the questionnaire was revised and the internal consistency was tested. The Cronbach’s alpha coefficient obtained 0.84 for the whole questionnaire and the values between 0.58 and 0.87 for different items of the questionnaire were arguments for the fidelity and validity of the questionnaire.

A 5-point Likert response scale was used to construct the questionnaire. On the 5-point Likert scale of the questionnaire for assessing conditions and barriers in the process of implementing the PBM standard, the scores represented the perception of the employees of the existence of specific actions that may support or hinder the implementation of PBM. Scores 1 (total disagreement) and 2 (disagreement) represented a negative perception of the provision of the necessary conditions for the implementation of PBM measures and of the attitudes and individual and group behavior patterns promoted at the unit level to determine adherence to the standard. Scores 4 (agreement) and 5 (total agreement) were obtained when perceptions were associated with an environment supportive of PBM implementation and were characterized by communications based on mutual trust, shared perceptions of the importance of safety, and confidence in the effectiveness of preventive measures. Score 3 was associated with neutral perceptions. These responses are interpreted as a lack of barriers in implementing PBM, but further efforts are needed to ensure that attitudes and behavioral patterns are associated with a culture of patient safety. Likert scale data were converted to percentages to facilitate a clearer understanding of the distribution of responses.

### 2.1. Study Population

The target population was the staff working in surgical and intensive care departments in participating Romanian hospitals. Participation was voluntary and the questionnaire was distributed to both public and private hospitals with surgical departments in Romania. A total of 113 units out of 497 public and private hospitals participated in this study. Participating hospitals were not randomized and were chosen based on their previous participation in similar surveys. A total of 98% of the enrolled hospitals were public and 2% were private hospitals. All enrolled hospitals include surgical wards and anesthesia and intensive care departments with similar organizations. However, 33% of the hospitals were university hospitals, 49% were city and municipal hospitals and 18% were county hospitals, so there may have been differences in their characteristics between academic and county or municipal hospitals in terms of guideline implementation.

All surgeons and anesthesiologists involved in the study had the capacity to identify and to prescribe anemia therapy in their departments.

The sample was pre-estimated considering the data on the situation of doctors in Romania according to the National Institute of Statistics [13] in 2022, data available at the time of planning the study. The number of hospitals with surgical wards where these doctors work was also taken into account. A total of 1100 questionnaires were distributed. Out of these, we obtained 486 completed questionnaires, but only 413 were valid. The target group was chosen so that we could obtain information from all surgical and intensive care unit (ICU) doctors who are competent to prescribe treatment for anemia. Taking into account the recommendations of the Guidelines for Patient Blood Management [14], the surgical specialties, in addition to intensive care, that we addressed in the study are the following: general surgery, cardiovascular surgery, urology, thoracic surgery, gynecology, neurosurgery, and orthopedics–traumatology. According to data from the National Institute of Statistics [13], in 2022 there were 3005 ICU physicians and 9063 surgeons working in Romania, according to the mentioned specialties. The data presented support the fact that ICU doctors represent 24.9% of the total number of doctors covered by the study at a national level. This numerical difference between ICU doctors and surgeons justifies the fact that surgeons responded in a higher proportion. On the other hand, the response rate of ICU physicians, representing 15.8% of the total study respondents, also indicates a lower responsiveness of ICU physicians to the request to complete the questionnaire.

### 2.2. Data Collection Procedure

Questionnaires were distributed electronically via the online platform Google Forms. Invited participants were provided with a letter of intent with details about the objectives of the study, the voluntary nature of participation, and assurances that the identity and confidentiality of respondents would be preserved. The invitation to complete the questionnaire and the online questionnaire were distributed from 19 April to 26 May 2023.

### 2.3. Inclusion and Exclusion Criteria

Anesthesiologists and surgeons working in enrolled hospitals were included in the study. The study excluded questionnaires with incomplete responses, those that only included the respondent’s demographic information, and those that provided the same answer to every question. Out of 486 completed questionnaires, 73 were excluded. A total of 413 final questionnaires, or 85% of the total, were included in the study.

### 2.4. Data Analysis

Data cleaning and statistical analyses were performed using IBM SPSS Statistics 27 for Windows. A *p*-value < 0.05 was considered statistically significant. Descriptive statistics were performed, and nominal variables were reported as frequencies and percentages. Data analysis was performed after checking all assumptions of normality. Data were interval-level and independent. Continuous variables were tested for normality using the Kolmogorov–Smirnov test, and descriptive data were reported using means and standard deviations or means and interquartile range, as appropriate. Pearson’s exploratory correlations assessed relationships between key items. A comparison was made between the two categories of doctors in terms of the degree of implementation of the standard. To compare the means of the two groups, a *t*-test for independent samples and Levene’s test for equality of variances were performed. The Pearson correlation coefficient was used to measure the strength and direction of the association between the level of PBM implementation and seniority on the ward.

## 3. Results

Of 413 questionnaires analyzed, 347 were answered by surgeons (84.2%), while 15.8% were anesthesia and intensive-care physicians. The greatest number of questionnaire answered by the surgeons as compared with anesthesiologists may be determined by the ratio of surgeons to anesthesiologists in enrolled hospitals, and increased by the willingness to respond of the surgeons. The highest percentage of respondents (24.7%) had 21 years or more of professional experience, followed by 23.1% of physicians with 11–15 years. The category most under-represented in the responders’ group were doctors with less than one year of professional experience (4.6%).

A total of 28.3% of the responders stated that they had less than one year of experience in the ward/department in which they were working at the time of completing the questionnaire (Table 1). A total of 55.7% of the participants had up to 10 years of experience in the ward/department where they were employed at the time of the survey.

A total of 5 of the questions were addressed to the specific actions of PBM, while 7 of the questions addressed the knowledge on PBM and organizational aspects making PBM possible (Figure 1 and Figure 2).

As it can be observed in Figure 2, the receptivity of health care staff was indicated by almost 90% of the responders. However, the lack of human resources, the first obstacle mentioned by 23.70% of the participants in the study (Figure 3), makes it difficult to involve staff in PBM. A total of 84.20% of the participants considered that they have enough information about the standard. Protocols and procedures on PBM in hospitals are issued and are accessible as related by 69.90% of respondents.

There was a positive and statistically significant correlation between the items assessing the degree of implementation of the PBM standard and seniority in the profession (*p* < 0.001). A correlation coefficient of 0.245 suggests a moderate positive relationship between the two variables. This significant positive correlation shows that individuals with more years of professional experience were more likely to be involved in implementing PBM practices in the hospital.

When PBM implementation was examined for surgeons and anesthesiologists, anesthesiologists had a greater level (73.91%) than surgeons (53.30%).

Figure 3 illustrates the factors that are considered an obstacle for the implementation of PBM in Romanian hospitals, as perceived by the respondents. Among the identified factors, the most frequently mentioned were “Lack of human resources—physicians” (23.7%), followed by “Lack of knowledge/information on PBM” with 16.9% and “Lack of PBM education of medical staff” with 16.2%.

Other notable factors include “Not convinced of the benefits of applying PBM” at 12.6% and “Lack of patient compliance” at 8.0%. Despite the fact that “Lack of knowledge/information about PBM” was mentioned as one of the priorities (16.9%) hindering the implementation of the standard, however, only 0.7% of the responders mentioned knowledge of the standard as a barrier. The explanation may consist of the difference between PBM as a concept and national regulations on the PBM standard, which may be not so well known as PBM.

Regarding the question on who should be more involved in order to successfully implement PBM, most respondents (35.7%) answered that anesthesiologists should be more involved in PBM implementation, followed by “Surgeons” with 23.6%, and “Hematologists”, mentioned by 18.4% of the respondents (Figure 4).

In response to the open question in the questionnaire asking for suggestions on the implementation of the PBM standard, we received specific comments in 64 questionnaires. A high proportion of the comments, 51% of the comments, were related to recommendations for the organization of trainings and debates on the PBM standard: “we need trainings”, “more information to get”, “to organize multidisciplinary debates on this topic”, “training sessions”, “to distribute information materials”, etc. Specific comments referred to the way procedures are developed in the hospital: “Procedure should be formulated simply, in progression YES/NO”, “Procedure should be practical”.

## 4. Discussion

Our study investigated the conditions of PBM implementation in Romanian hospitals and the barriers in implementing this standard. In Romania, since 2019, patient blood management should be implemented as a standard of care [14]. The largest effects on staff knowledge of this subject and on improving the detection and treatment of preoperative anemia are related to information and awareness of the PBM concept [15,16]. However, the implementation of PBM is still far behind expectations both in Romania [9] and in other countries where the standard has been regulated [17,18].

There are several barriers to transferring the PBM guidelines into practice, and identifying them is essential to coming up with implementation plans that consider the unique requirements of the medical staff as well as the environment in which they operate in each facility [19]. In addition to overcoming the barriers, specified in studies, to PBM implementation [14,20], promoting patient safety culture is also a challenge that can significantly contribute to improving patient care outcomes [21].

The analysis of respondents’ perceptions also provided a specific picture of the conditions that are provided in the hospitals enrolled and the barriers to the implementation of the standard. A total of 89.80% of the study participants were receptive to changes in order to implement PBM. Routine iron measuring in anemic patients was indicated by 81.11% of responders.

Nonetheless, recognizing the need for iron level measurement does not provide a clear answer as to who is responsible for implementing PBM. The responses to the question on who should be involved in implementing PBM (anesthesiologists 35.70%, surgeons 23.60%, and hematologists 18.40%) show that there is no consensus on which physician should be responsible for this. Other studies also indicate that physicians refer to different professionals they consider responsible for the treatment of a preoperative anemia and there is consensus on who is responsible only in a few hospitals [22]. All these data support the need to clarify how responsibilities are distributed at the hospital level by a key leader for managing the PBM project at the institutional level.

Despite this, only 23.50% of the participants answered that they carry out pre-anesthetic consultations in elective patients 7 days or more before surgery, as required by the regulations, and only 55.93% said they routinely administer iron and folic acid preoperatively if the patient is anemic.

Measures that are known to be relevant should be part of the routine of medical staff and integrated into the organizational culture [23]. Organizing a monitoring system at hospital level that is easy to use and provides real-time process information would impact on the daily work of staff and generate increased confidence in the impact of implementing the PBM standard, given that 12.60% stated that they are not convinced of the benefits of implementing PBM. Only 36.60% of respondents reported that the hospital’s PBM implementation committee ensures physician involvement in monitoring activities to address and treat preoperative anemia.

Continuous provision of feedback on the status of PBM implementation and presentation of impact data are strategies to motivate staff to become involved in the PBM project. Quarterly reports on the impact of the PBM project sent to the structures involved increase the level of adherence to PBM measures [24]. Integration into protocols and information may not generate the desired impact. Feedback should complement the hospital’s education program to ensure continued implementation.

To facilitate implementation, different strategies have been established and there is a major preoccupation to disseminate PBM to prescribing health professionals. The introduction of electronic patient record systems in many United Kingdom hospitals provides an opportunity to follow up blood ordering and reduce inappropriate transfusions [25]. Prospective monitoring of blood orders provides the opportunity to intervene to avoid unnecessary transfusions. Although a retrospective review is easier to achieve, it misses the opportunity to intervene to prevent inappropriate transfusions. Both methods of review are facilitated by the use of information technology. Warning “alerts” if the prescribing physician attempts to order a transfusion if the most recent laboratory tests do not fall within those recommended as transfusion initiating factors are a solution for control the process and monitor the implementation of PBMs. This would overcome barriers related to the attitude and behavior of medical staff. Another way to stimulate PBM implementation proposed in the studies is to modify guidelines at an institutional level. But these changes will be implemented if they are supported at the top management level [12]. An action that is promoted from the bottom up, although likely to be more acceptable, is slower.

The most relevant obstacle in implementing the standard was related to the lack of human resources. If the assessment of patient safety culture in 43% of the anesthesia and ICU structures in Romania [26] found that the dimension that needs the most improvement is staffing, this study also revealed that the lack of staff (doctors) is an obstacle in the perception of 23.70% of respondents, and is the most important barrier in implementing PBM.

Perceived human resource gaps create “organizational challenges”, “legal challenges” and “personal challenges” in health-and-safety standards implementation. Organizational challenges include limited financial resources, remuneration discrimination, staffing discrimination, workload imbalance, poor organizational coordination, ineffective cross-sectoral relationships, parallel decision-making, ineffective allocation of human resources, psychological distress, reduced self-confidence, burnout, increased workload, and low job satisfaction [27]. These are also motivational barriers to participation in training programs or adaptation to strategies and policies promoted by the health facility.

More than 69% of the responding physicians stated that “There is good communication between the surgeon and anesthesiologists to perform pre-anesthetic consultations on elective patients”; however, regulations for interdisciplinary cooperation for PBM implementation are still necessary [28]. Related to this, in our study there were also differences in perception of the degree of PBM implementation between anesthesiologists (69.4%) and surgeons (41.7%), similar to other studies in the literature [29].

The leader of the PBM project in the hospital must take on the role of building communication networks between physicians and the professionals involved. The key role of this leader is to ensure institution-wide coordination of education programs, documentation of processes and the creation of a common vision on how to implement PBM.

The results of our study showing that physicians with more years of professional experience are more likely to be involved in implementing PBM practices in the hospital are similar to those of other studies [30] assessing patient safety culture, which also support the fact that gaining experience is associated with greater involvement in promoting and sustaining patient safety culture. On the other hand, our results highlight the need for more knowledge and information on PBM, which was mentioned as the second barrier to implementing the standard by 16.90% of the study participants. This need comes in a e context in which 42.60% of the respondents mentioned that “Regular information on how to implement PBM is organized in the hospital” and less than 1% of the doctors stated that the PBM standard would not be known by doctors, a finding that was reported by other studies on the PBM standard [31,32], which highlights the lack of continuous, practically informed implementation. This finding is similar to the result of a study conducted in 7 hospitals in Europe, in which approximately 24% of respondents were unaware of a correlation between preoperative anemia and perioperative morbidity and mortality, 29% responded that they did not have enough information about the best way to treat iron deficiency anemia before surgery, and 5% said they “do nothing” [33]. Part of the problem may be the large number of tools and measures proposed by the standard, as well as the knowledge required to identify and apply them appropriately in current work. The literature [34,35,36] provides mostly generalized findings on the effect of implementation strategies and not data on the specific implementation actions taken that had the greatest impact. This results may justify the need for “Organized regular briefings in the hospital on how the PBM is implemented” mentioned by 57.40% of the survey participants.

An outcome of the accreditation process in hospitals [37] is to ensure a culture of patient safety and to regulate processes through procedures and protocols. All hospitals involved in our study have gone through the accreditation process and 69.90% of participants report that they have “access to protocols or procedures aimed at the treatment of pre-operative anemia”. However, in the open question in the questionnaire on respondents’ comments and suggestions it was mentioned that there is still a need for these protocols to be easier to understand and apply. If we talk about the more than 100 PBM measures that need to be part of the daily routine of physicians, there is a clear need to implement the simplified recommendations through working models organized by the PBM project coordinator in the hospital [38].

Only 12.12% of respondents mentioned the need for hospital management involvement in PBM implementation. If we look at it from the perspective of allocation of financial and human resources, the support of hospital managers is essential [39]. Ensuring sufficient and continuous resources is a requirement for surmounting organizational obstacles that might interfere with a PBM program. Beyond ensuring resources, organizational issues, including supervision and support of the PBM leader, monitoring of equipment and information systems, policies and procedures for clinical care, documentation, and response to noncompliance and adverse events, are also the responsibility of hospital management. These are either fulfilled or the need to ensure top management support is not recognized.

Our study has several limitations. First of all, the study included only 23% of hospitals with surgical departments. Although we received responses from a large number of physicians from surgery and anesthesia and intensive care units from public and private hospitals, there may be a selection bias in sending the questionnaire to hospitals that usually collaborate in such studies to ensure a high response rate. Similarly, hospitals may have been different in terms of guideline implementation; however, this was not an aspect targeted in our study, where we aimed for a more general identification of the barriers regardless of the organizational characteristics of surgical and anesthetic departments. Also, distributing the questionnaire by email and inviting respondents to complete it is a limitation because online surveys are relatively easily deleted. Another limitation of this research was the discrepancy between the number of surgeons and anesthesiologists answering the questionnaire, which was dominated by surgeons, proportionally reflecting the staff number in specialties. Additionally, the use of Likert-type scales may be another possible limitation of this study. Although Likert-type scales remain prevalent in health studies due to their simplicity and ease of use, response selection bias may alter the results. There is a tendency for respondents, due to desirability, to choose the “agree” option on the Likert questionnaire [40]. This conformity bias can potentially distort the results of the study and give a false positive picture of the actual attitudes and behaviors of individuals.

The present study should be complemented by the assessment of patients’ perceptions because the benefits of PBM are largely unknown to patients, even though they are the ones who stand to gain from PBM, with significantly improved clinical outcomes, safety, and reduced average length of hospitalization [41]. Lack of patient information and involvement in decision-making can also be considered a deficiency in quality of care and a violation of patient rights. Studies and programs targeting patients by educating and advocating for their rights could also contribute to raising awareness of patient blood management and stimulating PBM implementation.

## 5. Conclusions

In conclusion, the perception of surgeons and anesthesiologists of the implementation of the PBM in hospitals with surgical departments in Romania is supportive of this standard. Although the degree of implementation is higher among anesthesiologists, there is still a need for more investments in human resources in all professional categories, which needs to be supplemented and also developed through training programs to increase adherence to the PBM standard.

Future research should address the benefits and barriers of concrete implementation practices in different types of hospitals in order to provide feasible models for translating the evidence into practice that can be integrated into healthcare unit procedures and protocols.

## Figures and Tables

**Figure 1 healthcare-12-00760-f001:**
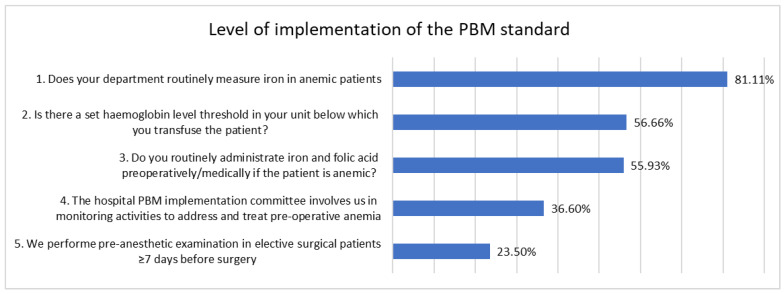
Participants’ perception of the degree of implementation of the PBM standard.

**Figure 2 healthcare-12-00760-f002:**
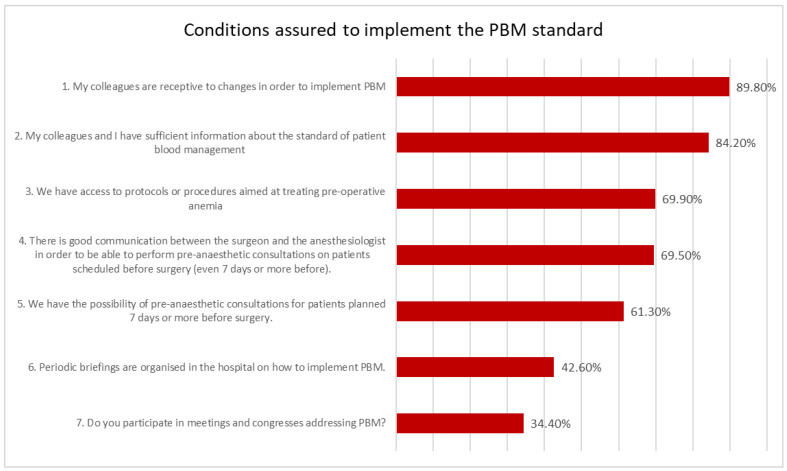
Perceptions of the participants of the study of the conditions provided at institutional level to promote the implementation of the PBM standard.

**Figure 3 healthcare-12-00760-f003:**
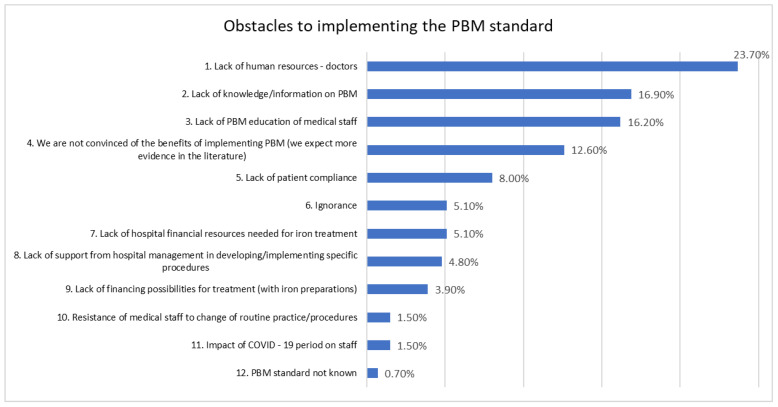
Principal obstacles to the PBM (Patient Blood Management) standard’s implementation in Romanian hospitals.

**Figure 4 healthcare-12-00760-f004:**
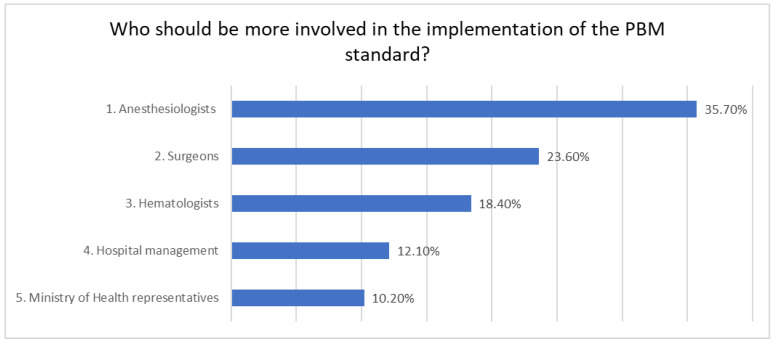
Perception of survey participants of who should be more involved in the implementation of the PBM standard.

**Table 1 healthcare-12-00760-t001:** Demographics of study participants (*n* = 413).

	*n* (%)
Professional categories	
Surgeons	347 (84.2)
Anesthesiologists and intensivists *	66 (15.8)
Professional experience (yrs)	
Less than a year	19 (4.6)
1–5 years	65 (15.7)
6–10 years	48 (11.6)
11–15 years	95 (23.1)
16–20 years	84 (20.3)
21 years or more	102 (24.7)
Years employed in the department	
Less than a year	117 (28.3)
1–5 years	57 (13.8)
6–10 years	56 (13.6)
11–15 years	47 (11.4)
16–20 years	75 (18.2)
21 years or more	61 (14.7)

* In Romania, anesthesia and intensive care are included in a single specialty.

## Data Availability

The data presented in this study are available on request from the corresponding author. The data are not publicly available due to privacy and ethical reasons.

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
