# Peer review of "Perceptions of the Conditions and Barriers in Implementing the Patient Blood Management Standard by Anesthesiologists and Surgeons"

_healthcare, 2024, doi:10.3390/healthcare12070760_

Round 1

Reviewer 1 Report

Comments and Suggestions for Authors

I read with interest the manuscript by Kazamer et al. on the perception of the conditions and barriers in applying the patient blood management standards by anesthesiologists and surgeons. The work is sound and well written. I have only minor comments to improve the manuscript:

- Introduction is quite long. Please summarize it in order to provide a brief background for the study.

- Line 131-133. There is no need to specify how percentages are calculated.

- Did the study receive an approval from local ethical committee? Please provide code and date of approval.

Author Response

Dear reviewer,

On behalf of our research group, we would like to thank you for your time and your comments. We highly appreciated your recommendations and taking into consideration your suggestions, we made the following changes to the manuscript:

  • The introduction was reduced as suggested. We hope that now it is easier to follow and offers enough background for the study.
  • The code and date of approval by the Ethical Committee was added in the dedicated section, lines 426-428.

We are hopeful that the quality of the manuscript has been improved and that we fulfilled your requirements. Thank you very much for taking into consideration the publishing of our manuscript. 

Reviewer 2 Report

Comments and Suggestions for Authors

The authors present a study about patient blood management practices in Romanian hospitals. The work is highly interesting and needed for improving the establishment of such standards in medical care worldwide. I suggest minor revision, since only small points are to be mended.

11.      Page 2 line 55 reference missing.

22.      Page 2 line 59 reference missing.

33.      Page 2 line 67 reference missing

44.      Page 2 line 69 reference missing

55.      Page 2 line 75 reference missing.

66.      Page 2 line 78 reference missing.

77.      Page 2 line 81 reference missing.

88.      Page 2 line 86 reference missing.

99.      Page 2 line 98 reference missing.

110.   department/department“

111.   The authors could cite also international standards. https://www.aabb.org/aabb-store/product/standards-for-a-patient-blood-management-program-3rd-edition-print-14892454

112.   Nowadays and in the future there are non-blood based blood replacement systems and in future there are hemoglobin particle carriers,1 which can already be upscaled.2 The authors should also write a few sentences about such systems which could also be administered to patients.

References

(1)        Duan, L.; Yan, X.; Wang, A.; Jia, Y.; Li, J. Highly Loaded Hemoglobin Spheres as Promising Arti Fi Cial Oxygen Carriers. ACS Nano 2012, 6 (8), 6897–6904.

(2)        Li, W.; Gai, M.; Rutkowski, S.; He, W.; Meng, S.; Gorin, D.; Dai, L.; He, Q.; Frueh, J. An Automated Device for Layer-by-Layer Coating of Dispersed Superparamagnetic Nanoparticle Templates. Colloid J. 2018, 80 (6), 648–659. https://doi.org/10.1134/S1061933X18060078.

Author Response

Dear reviewer,

On behalf of our research group, we would like to thank you for your time and your comments. We highly appreciated your recommendations and taking into consideration your suggestions, we made the following changes to the manuscript:

  • Regarding the missing references from the introduction, this section was reduced as one of the reviewers recommended and we hope that now all references are in place.
  • The error regarding “department/department“ was corrected on line 195
  • International standards (https://www.aabb.org/aabb-store/product/standards-for-a-patient-blood-management-program-3rd-edition-print-14892454) were cited in the manuscript on line 43
  • New information about hemoglobin particle carriers was added on lines 46-51.

We are hopeful that the quality of the manuscript has been improved and that we fulfilled your requirements. Thank you very much for taking into consideration the publishing of our manuscript. 

Reviewer 3 Report

Comments and Suggestions for Authors

dear author.

Thank you for your work. I found it interesting to read

I would like to make a series of clarifications with the intention of improving the scientific quality of it.

- Was the study submitted to the previous analysis by an ethics committee? Have the guidelines of the Helsinki declaration been taken into account?

- Did the selection of the hospitals respond to any type of randomization criterion? did all the selected centers share the same characteristics?

- The number of participants, the calculation of the necessary sample, was calculated beforehand? What criteria were used to select this number of participants and not others?

- Was the questionnaire sent to all physicians with the capacity to order anemia therapy in intensive care units? Why are most of the responding professionals surgeons?

- Does the questionnaire used have any kind of validity? Is it a validated questionnaire?

- The presentation of the graphs should conform to the same format.

Thank you for your efforts

Author Response

Dear reviewer,

On behalf of our research group, we would like to thank you for your time and your comments. We highly appreciated your recommendations and taking into consideration your suggestions, we made the following changes to the manuscript:

  • The code and date of approval of the Ethics Committee as well as the statement regarding the guidelines of the Helsinki Declaration were added on lines 426-428.
  • Regarding the characteristics of the selected centers, all enrolled hospitals include surgical wards and anesthesia and intensive care departments with similar organization. However, 33% of the hospitals were university hospitals, 49% were city and municipal hospitals and 18% were county hospitals, so there may have been differences in their characteristics between academic and county or municipal hospitals in terms of guidelines implementation (lines 131-135).
  • Regarding the calculation of the necessary sample, was pre-estimated considering the data on the situation of doctors in Romania according to the National Institute of Statistics in 2022, data available at the time of planning the study. The number of hospitals with surgical wards where these doctors work was also taken into account. A total of 1100 questionnaires were distributed. Out of these, we obtained 486 completed questionnaires, but only 413 were valid. The target group was chosen so that we could obtain information from all surgical and intensive care units doctors who are competent to prescribe treatment for anemia (lines 138-144).
  • We added an explanation why most of the responding professionals are surgeons on lines 144-155.
  • All surgeons and anesthesiologists involved in the study had the capacity to identify and to prescribe anemia therapy in their departments – lines 136-137.
  • Regarding the validation of the questionnaire, five experts in PBM standard implementation pretested and reviewed the prepared questionnaire for accuracy and clarity. Following the recommendations the questionnaire was revised and the internal consistency was tested. The Cronbach's alpha coefficient obtained of 0.84 for the whole questionnaire and the values between 0.58 and 0.87 for different items of the questionnaire were arguments for the fidelity and validity of the questionnaire (lines 102-107).
  • The figures were modified to share the same format (lines 202, 204, 223, 242)

We are hopeful that the quality of the manuscript has been improved and that we fulfilled your requirements. Thank you very much for taking into consideration the publishing of our manuscript. 

Round 2

Reviewer 3 Report

Comments and Suggestions for Authors

dear authors

thank you for taking into consideration the comments made.

I consider them sufficient to improve the quality of the research report presented.

Greetings